# Tele-Assisted Behavioral Intervention for Families with Children with Autism Spectrum Disorders: A Randomized Control Trial

**DOI:** 10.3390/brainsci10090649

**Published:** 2020-09-18

**Authors:** Flavia Marino, Paola Chilà, Chiara Failla, Ilaria Crimi, Roberta Minutoli, Alfio Puglisi, Antonino Andrea Arnao, Gennaro Tartarisco, Liliana Ruta, David Vagni, Giovanni Pioggia

**Affiliations:** Institute for Biomedical Research and Innovation (IRIB), National Research Council of Italy (CNR), 98164 Messina, Italy; flavia.marino@cnr.it (F.M.); paola.chila@irib.cnr.it (P.C.); chiara.failla@irib.cnr.it (C.F.); ilaria.crimi@irib.cnr.it (I.C.); roberta.minutoli@irib.cnr.it (R.M.); alfio.puglisi@irib.cnr.it (A.P.); antoninoandrea.arnao@cnr.it (A.A.A.); gennaro.tartarisco@cnr.it (G.T.); liliana.ruta@cnr.it (L.R.); david.vagni@cnr.it (D.V.)

**Keywords:** telehealth, ASD, ABA, behavioral intervention, RCT

## Abstract

Background: Telehealth is useful for both autism spectrum disorder (ASD) diagnosis and treatment, but studies with a direct comparison between teletherapy and traditional in-person therapy are limited. Methods: This randomized control trial—ISRCTN (International Standard Randomised Controlled Trial Number) primary clinical trial registry ID ISRCTN15312724—was aimed at comparing the effect of a tele-assisted and in-person intervention based on a behavioral intervention protocol for families with children affected by ASDs. Forty-two parents with children with autism (30 months to 10 years old) were randomly assigned to 12 sessions of an applied behavioral analysis (ABA) intervention implemented in an individual and group setting, either with or without the inclusion of tele-assistance. Pre- and postintervention assessments were conducted using the Home Situation Questionnaire (HSQ-ASD) and the Parental Stress Index (PSI/SF). Results: Substantial improvements in the perception and management of children’s behavior by parents, as well as in the influence of a reduction in parent stress levels on said children’s behavior through the use of a tele-assisted intervention, were obtained. Conclusions: This randomized controlled trial demonstrates the evidence-based potential for telehealth to improve treatment of ASDs.

## 1. Introduction

Social communication deficits and restricted and repetitive interests and behaviors are the core symptom domains of autism spectrum disorders (ASDs) [1,2]. In addition, language, perceptual and sensory processing abnormalities, as well as delays in ASD detection and access to care during the crucial toddler years, remain unique challenges [3,4]. Nevertheless, intensive evidence-based treatments supported by parent-mediated interventions are crucial to obtain improvements in the developmental trajectories and functional outcomes of children with ASDs [5]. Telehealth represents a new technology able to emphasize the strengths of current treatment methods [4,6], mainly because it offers: (i) possible remote diagnostic methods with a consequent reduction in the delay in ASD detection; (ii) a continuous and ubiquitary access to care for families with children with autism; (iii) the chance to directly involve family members in the child’s development by actively applying effective parent-mediated interventions.

Telehealth is defined by the Global Health Observatory (GHO) of the WHO as the delivery of healthcare services, where patients and providers are separated by distance [7]. Information and communication technologies are used to interact directly with both clinicians and algorithms. Telehealth is also used for exchanging information to enable the diagnosis and treatment of diseases and injuries, as well as for improving patient access to quality, cost-effective, health services, wherever they may be. It is especially relevant for vulnerable groups [7].

Telehealth is a delivery model demonstrating the potential to deliver early intervention services effectively and efficiently, thereby improving access and reducing the impact of resources shortages in underserved areas. The use of a telehealth delivery model facilitates interdisciplinarity and services coordination, and it also makes possible the consultation with specialists not available within a local community [8]. 

Despite the undeniable advantages of telehealth and the interest in exploring the feasibility of implementing telehealth-supported behavioral interventions, the number of papers suggesting its effectiveness with children or adults with ASDs are extremely limited in the literature [3,6,9,10,11,12]. Recent findings of a scoping review [3] show the potential for telehealth to improve access for the assessment and diagnosis of ASDs, even in the very early diagnostic stages. A recent systematic literature review analyzed 28 studies reporting how telehealth is useful for both diagnosis and treatment in the case of ASDs [9], and the authors concluded that more research is needed before considering telehealth as an efficacious evidence-based treatment model. This seems mainly due to the shortage of direct comparisons between on-site and online teletherapy with ASDs [6]. The most widely used implementation is parent training programs for families with children with ASDs in order to increase their knowledge about the condition, to suggest behavioral intervention strategies to manage the children’s inadequate behaviors in everyday life, and to recommend psychotherapy for parents to ameliorate their emotional burden and to reduce their stress levels. In various studies, the objective of telehealth has been to make parents competent, as well as to ensure that they are able to perform a functional analysis of their children’s behavior and to learn functional communication techniques. Such objectives are usually achieved through dedicated remote training with the support of educational videos, web-based programs, and weekly videoconferencing coaching sessions with an operator [3,9,10,11,12]. The results show that, in the case of direct training with a behavioral analyst, parents become competent and increase their ability to perform a functional analysis and to use functional communication techniques. Thus, telematic tools become effective, acceptable, and usable for the parents [3,6,9,12,13,14,15,16,17]. The application of these techniques by parents makes it possible to reduce the inadequate behavior of children, with a peak reduction greater than 90%, and to improve the children’s social communication skills. 

## 2. Materials and Methods

To contribute to the growing research of telehealth applied to ASDs, in this paper, we aimed to investigate the feasibility and efficacy of a tele-assisted parent-mediated intervention for children with ASD in the context of an applied behavior analysis (ABA)-based treatment [9,18]. We compared, in a randomized controlled trial (RCT), the efficacy of the parent-mediated intervention that was delivered via telehealth or in-person approaches. The RCT is registered with ID ISRCTN15312724 at the ISRCTN (International Standard Randomised Controlled Trial Number) primary clinical trial registry—http://www.isrctn.com/ISRCTN15312724. Using tele-interventions, we aim at reducing parental distress due to commute time, strict schedules and unfamiliar environments. Furthermore, we aim to create a higher family engagement in the therapy and foster skills generalization in the home context.

We hypothesize that the parents of children on the autism spectrum randomly assigned to the tele-assisted intervention will perceive (1) a decrease in the severity of disruptive and noncompliant behavior in their children after the intervention and (2) that their children become easier to manage compared to children on the autism spectrum who undergo the intervention without telehealth assistance, and that he/she will show (3) lower parental distress and (4) improvements in the parent–child functional interaction.

The tele-assisted intervention was designed and implemented through a web platform, providing video conference tools and ABA assignments for parents that included cues, prompts, and reinforcements. The tele-assisted intervention was planned by chartered ABA psychotherapists, well-experienced in parent coaching and treatment, who delivered the intervention to both groups. The psychotherapists were teamed with bioengineers who implemented the intervention protocol via a web platform. The web platform was developed within the G Suite, a suite of cloud computing, productivity, and collaboration tools, software, and products developed by Google (https://gsuite.google.com).

### 2.1. Applied Behavior Analysis Therapy for Autism Spectrum Disorders

The ABA method studies particular dynamic interactions between the organism and its environment, and this method has been adapted to improve challenging behaviors in children with ASDs [18,19]. ABA therapy for autism is usually administered with a high intensity and used to achieve specific, measurable goals [19]. In this field, ABA is based on behaviorist theories that state that simple and complex behavior can be taught through a system of rewards and consequences. Most of the time, this therapy is intended to “extinguish” undesirable behaviors and to teach desired behaviors and skills.

ABA focuses on a behavioral approach, making it possible to simultaneously improve behavioral, cognitive, social, and communicative skills [19,20]. This system uses “reinforcement” (i.e., rewards) to motivate children with autism to learn new skills, as well as multiple trials that start with a prompt (i.e., antecedent) to execute the desired behavior. ABA therapy starts with an evaluation to determine a child’s challenges and strengths in the areas of behavior, cognition, communication, and social interactions. Then, the ABA therapist sets appropriate goals for the child and recommends a particular number of hours of therapy per week.

The basic structure of an ABA intervention is a set of repeated behavioral trials consisting of an antecedent, behavior, and consequence in discrete trial training (DTT) implementation, while in natural environment teaching (NET), the motivators are selected by the preferences of the children during natural social interactions, while any attempt at compliance is rewarded [21].

Overall, systematic reviews [18,19,21,22] have demonstrated that ABA interventions show promising evidence in terms of efficacy and are a viable intervention for individuals on the autism spectrum, both in telehealth and in-person [9,23], although the variability of the study designs, the heterogeneity of the participants’ clinical presentation, and the methodological limitations reduce the generalizability of study findings. 

ABA protocols in children and adults on the autism spectrum are individually delivered, with the involvement of family members or in a group-based format. While individual ABA protocols ensure a better personalization of the therapy objectives and strategies to the specific needs and skills of the single subject, group-based ABA interventions take advantage of the influence of social interaction, promoting experience-sharing, improving self-acceptance, and supporting insights of both personal strengths and impairments.

### 2.2. Inclusion Criteria

The inclusion criteria were as follows: (1) parents of children aged between 30 months and 10 years; (2) a clinical diagnosis of an ASD for the children of the recruited families based on the Diagnostic and Statistical Manual of Mental Disorders-Fifth Edition (DSM-5) criteria from a licensed clinical child neuropsychiatrist; (3) DSM-5 severity scores from moderate (level 2) to severe (level 3) in both the social communication and the restricted interests and repetitive behaviors domains; (4) not being on psychiatric medication; (5) not receiving any other intervention directly related to behavioral skills during the trial.

All children had a previous diagnosis that was further confirmed through the assessment and the consensus of the experienced professionals on the research team (i.e., a child neuropsychiatrist and a clinical psychologist). 

### 2.3. Participants

Families were recruited as part of an ongoing research program and were tested at our clinical facilities. We enrolled *N* = 88 parents of 44 children with ASDs, aged 30 months–10 years. A first screening based on inclusion criteria was implemented, and *n* = 74 parents of children with ASDs were eligible; *n* = 36 (30:6 male/female) children fully met the entry criteria and their parents were enrolled in the present study (Figure 1).

Parents were randomly assigned to the tele-assisted group (TG) or to the control group (CG), applying exactly the same protocol without telehealth assistance. A randomized block design was used to ensure that intervention groups were balanced with respect to gender, age, and developmental quotient (DQ). Finally, another 30 parents were excluded due to missing data. There were no dropouts during the interventions. Data were complete for *N* = 20 parents (9:11 males/females) of *N* = 11 children (9:2 males/females; mean age in months = 69.6; standard deviation (SD) = 32.9; mean DQ = 68.8; SD = 21.4) in the CG and for *N* = 22 parents (10:12 males/females) of *N* = 12 children (10:2 males/females; mean age in months = 69.1; SD = 22.5; mean DQ = 63.8; SD = 16.9) in the experimental group (Table 1).

All of the above-mentioned children were attending mainstream public schools for 27 h a week, with a special teacher for 10–12 h. All children were Italian.

All children were scored at or above the clinical cut-off on the Autism Diagnostic Observation Schedule, Second Edition (ADOS-2), module three. The child psychologist collected information from parents concerning developmental milestones (including joint attention, social interaction, pretend play, and repetitive behaviors, with an onset prior to 3 years of age) and current behaviors.

### 2.4. Intervention Protocols

The protocol consisted of three consecutive phases. In phase I, all of the enrolled parents received 12 2 h-long plenary sessions of informative parent training about ASD characteristics and ABA/behavioral principles. The intervention protocol then consisted of two consecutive sections of 12 weeks each, i.e., phases II and III. The present study reports the results of the phase III comparison between CG and TG groups.

Phase II lasted 12 weeks. In phase II, all of the enrolled parents received 2 h/week of group behavioral therapy administered in homogeneous groups (based on the developmental age, target behaviors, and ASD level of their children). In this phase, all of the children of the enrolled parents received 1 h/week of one-to-one ABA therapy, where parents were allowed and invited to observe the therapists during treatment sessions.

Phase III lasted 12 weeks. In phase III, the intervention protocol consisted of the administration of 2 h/week of tele-assisted one-to-one behavioral parent training and coaching to participants belonging to the TG, while 2 h/week of in-person one-to-one behavioral parent training and couching was administered to participants belonging to the CG.

All of the therapies were administered by a clinical psychologist with a postmaster’s degree in behavioral modification and analysis. Considering the usual ABA protocols last 25–40 h/week, our protocol can be considered to be low intensity. Testing the efficacy of low-intensity protocols, mediated by parents in natural environments, is of the utmost importance to develop more efficient implementations. 

### 2.5. Ethics

All subjects gave their informed consent for inclusion before they participated in the study. The study was conducted in accordance with the Declaration of Helsinki, and the protocol was approved by the Ethic Committee of the Research Ethics and Bioethics Committee (http://www.cnr.it/ethics) of the National Research Council of Italy (CNR) (Prot. N. CNR-AMMCEN 54444/2018 01/08/2018). All of the parents of the children who took part in the study gave their consent to participate in this study, signing a written consent form.

### 2.6. Outcome Measures

The outcome measures for all of the TG and CG participants were assessed during the weeks before and after the intervention sessions (week 1 and week 12 of phase III, respectively). The general measures were assessed by direct observations of the parents. The primary outcome measurement tools were the Home Situation Questionnaire (HSQ-ASD) [24] and the Parental Stress Index (PSI/SF) [25,26], both of which are objective measures of the perception and influence of children’s behavior on the psychological state of their parents. The investigators who assessed the outcome measures were blinded to intervention allocation. There were no significant differences between the prephase III outcome measures of either group (Table 1).

#### 2.6.1. Home Situation Questionnaire (HSQ-ASD)

The HSQ-ASD is a caregiver-rated scale designed to assess the severity of disruptive and noncompliant behavior in children. Its modified and revised version for ASD consists of 27 items describing different situations or settings that are common for children on the spectrum. Parents are asked to indicate whether their children have problems with compliance in these situations and, if so, to rate the severity on a 0–9 Likert scale, with higher scores indicating greater non-compliance. Factor analysis of the questionnaire yielded two distinctive 12-items subscales: (1) Social Inflexibility (SI) (*α* = 0.84) and (2) Demand-Specific (DS) (*α* = 0.89) [24]. The first subscale comprises items regarding compliance with changes in daily social routines, while the second one is related to demand avoidance for daily living tasks. The two subscales are moderately correlated (*r* = 0.51). The subscale totals’ test–retest reliability were all significant with *r* = 0.57 for socially inflexible, *r* = 0.58 for demand specific, and *r* = 0.57 for the combined total. Convergent validity was assessed with previously known scales. HSQ-ASD was correlated with scales measuring problem behaviors and daily living skills, while was not correlated with IQ or communication skills.

#### 2.6.2. Parental Stress Index/Short Form (PSI/SF)

The PSI/SF is a self-assessment questionnaire designed for the early identification of factors that can compromise the normal development of a child. The Italian validation of the test affects only the short form (PSI/SF), which derives directly from the extended form, since it contains all entries with identical words. The test is based on the hypothesis that the stress that a parent experiences is the joint result of certain characteristics of their children, the parents themselves, and a series of situations closely related to their parental role. Data relating to the mother have been added to the short form, including age, marital status, education, and profession. This test investigates three main domains of stressors, namely, those associated with the characteristics of the children, those of the parents, and those of situational-demographic events. The short form is composed of 36 items, divided into three subscales: (1) Parental Distress (PD), which taps into parental feelings; (2) Parent–Child Dysfunctional Interaction (P–CDI), which focuses on the perception of the child as not responding to parental expectations; (3) Difficult Child (DC), which is centered on some of the characteristics of the child that make it easy or difficult to manage. The expected time to complete the questionnaire was between 10 and 15 min [25]. Raw total scores above 90 or 33 on the PD and DC subscales and above 27 on the PCDI subscale are considered clinically elevated. Test–retest reliability coefficients of the total stress score have been reported to be *r* = 0.84, for the PD subscale *r* = 0.85, for the PCDI subscale *r* = 0.68 and for the DC subscale *r* = 0.78. For the internal consistency of the PSI/SF, reports for total stress have been *α* = 0.91, for PD *α* = 0.87, for PCDI *α* = 0.80 and for the DC subscale *α* = 0.85. A subsequent study on parents of ASD children [26] found a similar internal consistency: PD *α* = 0.91, for PCDI *α* = 0.85 and for the DC subscale *α* = 0.82 and *α* = 0.91 for the total score.

### 2.7. Statistical Analysis

After having controlled for multivariate analysis of covariance (MANCOVA) assumptions of normality using the Shapiro–Wilk test and homogeneity using Box’s M test of the equality of covariance matrices, parametric statistics were applied in order to analyze the group effects on the intervention. Levene’s test of equality of error variances was performed post hoc.

Total HSQ-ASD and PSI/SF scores were used as primary outcome measures, while the single subscales were used as secondary measures. Group and parental gender were used as factors, while child age and DQ, together with the preintervention variables, were used as covariates. A two-sided test with an alpha level of 0.05 was used, after adjustment using the Šidák correction for multiple comparisons. Sensitivity analysis was performed using G*Power 3.1 for MANOVA with 2 groups and 2 factors and 4 covariates. With a total sample size of 42, we can identify large effects with sizes with *f*^2^(U) > 0.631. To increase the confidence in the results, we computed 95% bias-corrected confidence intervals for our data using a bias-corrected and accelerated (BCa) bootstrap (*n* = 1000).

Finally, to further explore the intervention effects, as an ancillary analysis we also used a paired t-test between pre- and postintervention variables for each group separately.

The raw data for each participant comprising the demographics and assessment, HSQ-ASD and PSI/SF preintervention and postintervention scores (Appendix A), as well as the complete statistics for the assumptions (Appendix A), are provided in the Appendix A, together with the complete analysis of covariance (ANCOVA) for the secondary outcomes variables (Appendix A), in order to provide an analytical picture of the sample distribution and to allow replicability.

SPSS software (v. 26, IBM Corporation, Armonk, NY, USA) was used to run statistical analyses.

## 3. Results

The demographic and clinical characteristics of the sample are reported in Table 1. No significant differences between the TG and the CG were found with respect to any demographic or clinical variable (Table 1).

The multivariate test found no effect of age, DQ, or parental gender on the outcome variables. There was a statistically significant effect in the outcome variables PSI/SF, *F*(2, 33) = 39.9, *p* < 0.001 (*Wilk’s Λ* = 0.293, *η*_p_^2^ = 0.707) and HSQ-ASD, *F*(2, 33) = 3.59, *p* = 0.039 (*Wilk’s Λ* = 0.821, *η*_p_^2^ = 0.179). There was also a significant effect of the experimental group, *F*(2, 33) = 11.7, *p* < 0.001 (*Wilk’s Λ* = 0.586, *η*_p_^2^ = 0.414) (Table 2). Univariate analyses led to a significant effect of the group on both outcome variables: PSI/SF (*M* = 8.28, standard error (*SE*) = 1.91; *F*(1, 34) = 18.7, *p* < 0.001, *η*_p_^2^ = 0.355) and HSQ-ASD (*M* = 0.742, *SE* = 0.358; *F*(1, 34) = 4.30, *p* = 0.046, *η*_p_^2^ = 0.112) with a larger decrease in the TG (Table 3).

Extending the analyses to the subscales, we found no interaction among the secondary outcome variables and age, DQ, or parental gender. All subscales had a significant multivariate effect (Appendix A). Univariate effects for group differences were significant for PD (*M* = 4.93, *SE* = 2.16; *F*(1, 31) = 5.23, *p* = 0.029, *η*_p_^2^ = 0.145), SI (*M* = 4.92, *SE* = 0.386; *F*(1, 31) = 5.69, *p* = 0.023, *η*_p_^2^ = 0.155), and DS (*M* = 1.16, *SE* = 0.383; *F*(1, 31) = 9.23, *p* = 0.005, *η*_p_^2^ = 0.229), while barely significant for P–CDI (*M* = 3.82, *SE* = 1.92; *F*(1, 31) = 3.94, *p* = 0.056, *η*_p_^2^ = 0.113) and not significant for DC (*M* = 0.173, *SE* = 2.63; *F*(1, 31) = 0.004, *p* = 0. 948, *η*_p_^2^ < 0.001).

Finally, we analyzed the two groups separately; as shown in Figure 2 (and Table 4), the TG displayed a significant improvement in PSI/SF, *t*(21) = 5.10, *p* = 0.001, with a decrease in stress of 7%. Conversely, the CG showed no significant change—*t*(19) = −1.19, *p* = 0.241. Likewise, the postintervention total scores of the HSQ-ASD decreased by 20% in the TG (Figure 3). The paired *t*-test indicated a significant improvement in the TG, *t*(21) = 2.32, *p* = 0.035, but not in the CG, *t*(19) = 0.145, *p* = 0.890. All children in the TG improved. Complete descriptive statistics of outcome variables are reported in the Appendix A.

The complete raw data (Appendix A) and pre- and postphase III results for each participant are reported in the Appendix A.

## 4. Discussion

In this study, the feasibility and efficacy of a tele-assisted ABA intervention program for families with children with ASDs was described and evaluated in a randomized controlled trial. To the best of our knowledge, this study is one of the first to focus on a tele-assisted ABA intervention program [9]. The intervention consisted of informative group parent training, behavioral group therapy, and individual therapy, engaging parents in behavioral interactive play activities with their young children, aged 3–10, with or without the addition of tele-assistance.

We found an effect of the tele-assisted intervention group on the parents’ stress levels and perception of the disruptive and noncompliant behavior of their children. This can be inferred by a significant change in the PSI/SF scores in the TG with a mean decrease of 8.28 points (−8.39%), in comparison to the CG and HSQ-ASD scores by the TG with a mean decrease of 0.742 points (−28.1%).

We found an increased ability of parents belonging to the TG to face stress, as highlighted by the PD subscale (−4.93 points; −16.1%), in comparison to the CG, and to cope with their children’s inadequate behaviors, as pointed out by a significant decrease in the SI (−0.920 points; −30.1%) and DS (−1.16 points; −49.5%) subscales and as suggested by the barely significant effect of the P–CDI subscale (−3.82 points; −14.2%).

Interestingly, we found no change in the judgment of the difficulties associated with a specific child (DC subscale, as well as if both the parental stress was reduced and the children’s behaviors were perceived as improved). This result is not surprising, given that our experimental intervention focused on teaching parents coping and interactional skills to manage challenging behaviors, rather than to change their perception regarding their children’s temperament. The strength of this study is also its limitation. Focusing on parental perception allowed us to better understand it, but in order to generalize the results and to assess the therapeutic efficacy of the protocol, future replication should also assess behavioral changes through direct and independent measures.

We had no parents drop out during the trial. All of the parents in the TG demonstrated high interest in the web-based platform and sustained motivation throughout each experimental session. Unfortunately, 40% of the parents did not complete all of the pre- and post-tests and were therefore excluded; thus, increasing the parents’ collaboration not only for the therapy sessions, but also to complete the tests would be important for future replications. Participants identified several benefits associated with telehealth including its flexibility, reduction in commute time, access to providers, and more family engagement. Parents and children experiencing the tele-assisted sessions showed the usually demonstrated parent–child interactions at home, especially initiating and sharing play activities. This is in line with the previous literature reporting how effective, acceptable, and usable ubiquitous communication technology is for the parents [27,28,29]. Moreover, as reported by themselves, all parents in the TG spontaneously practiced the learned lessons at home subsequent to the tele-assisted intervention, just as is usually reported by parents following in-person parent coaching sessions. 

Parent–child interactions were not influenced by computer, tablet, or general technology concerns, e.g., internet connection, digital divide, or anxiety regarding their performance when using technology. These qualitative results are in contrast with those reported by Cole et al. who identified access to high speed internet and the opinion that telehealth was not as effective as in-person treatment as their primary barriers [30]. Our hypothesis is that given the specificity of our center, which is highly focused on technology, recruited parents could had a more favorable propensity toward technology, and future study should replicate the results with a more diverse population and aim to validate structured interviews or questionnaires to assess parental propensity to initiate a tele-intervention in an easily replicable fashion.

Furthermore, while ABA interventions are usually time-intensive, we administered a low-dose intervention, suggesting that the additional hour of tele-assistance could be pivotal for the success of the implementation. From informal exchanges with the participants, many parents reported challenges in generalizing and personalizing the techniques learned during the courses and in generalizing the daily life situations the skills learned through observations in the laboratory. The addition of tele-assistance helped the therapists fill the gap, reaching families of children with ASDs in their homes.

This evidence encourages further experimental studies regarding the long-term application of tele-assisted interventions in ASD therapy.

## 5. Conclusions

The obtained results support the idea that an ABA tele-assistive intervention can be an effective treatment for families with children on the autism spectrum following initial parent training. In line with our hypothesis, we found a significant positive effect of the ABA tele-assistive intervention in terms of parents’ stress levels, perception of the disruptive and noncompliant behavior of their children, coping with their children in cases of inadequate behaviors, as well as in the influence on their children’s behavior. We expect that further studies with larger samples may replicate these findings. All of the raw data gathered, and the analysis described in this study, are reported in the Appendix A for future comparisons.

## Figures and Tables

**Figure 1 brainsci-10-00649-f001:**
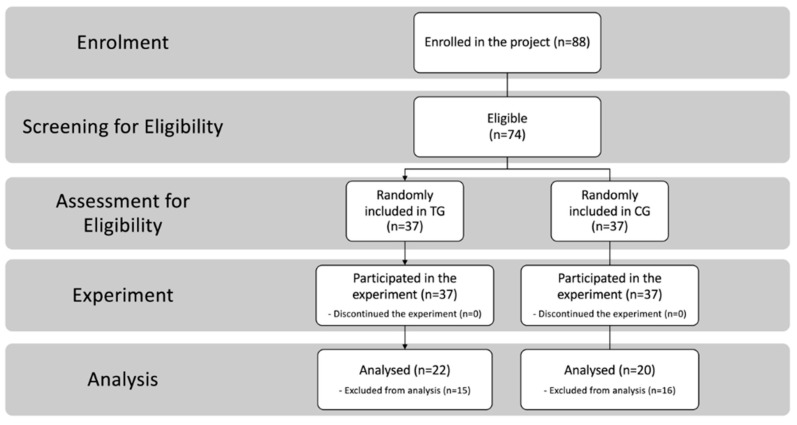
Subject recruitment, assignment, and assessment procedures.

**Figure 2 brainsci-10-00649-f002:**
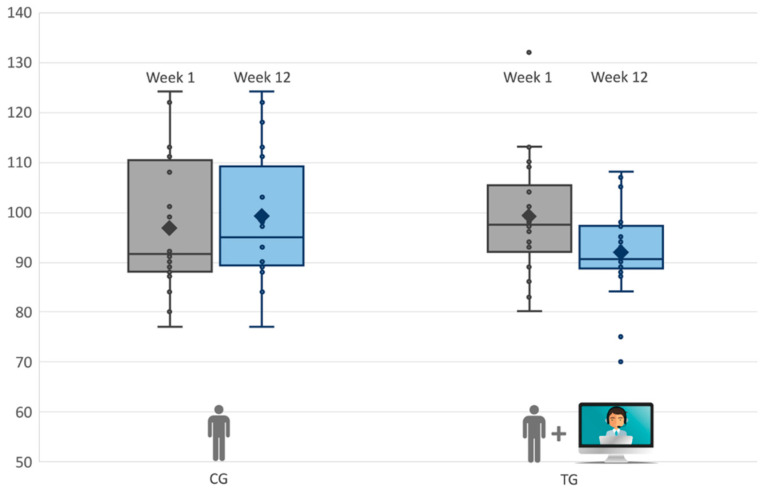
Box-plot of postphase III gains for Control Group (CG) and Tele-assisted Group (TG) in the Parental Stress Index/Short Form (PSI/SF) Test.

**Figure 3 brainsci-10-00649-f003:**
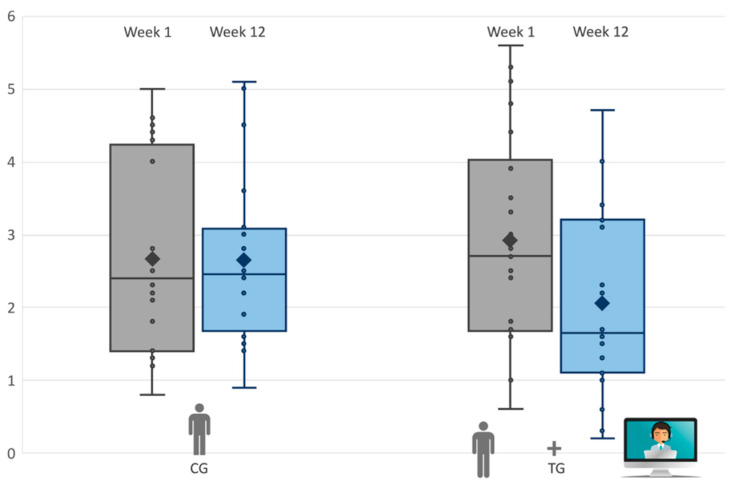
Box-plot of postphase III gains for gains for control group (CG) and tele-assisted group (TG) in the Home Situation Questionnaire-ASD Version (HSQ-ASD) Test.

**Table 1 brainsci-10-00649-t001:** Demographic and clinical characteristics of the sample.

			CG (*n* = 20)		TG (*n* = 22)		Comparison between Groups				
Preintervention Variables			*M*	*SD*	*M*	*SD*	*M*	*SE*	*t*	d.f. *	*p*-Value
Child Demographic	Age in months		69.6	32.9	69.1	22.5	0.509	8.79	0.058	33.1	0.945
	DQ		68.8	21.4	63.8	16.9	4.93	5.99	0.822	36.1	0.416
Outcome Variables	*PSI/SF*	PD	30.1	9.31	31.8	10.5	−1.77	3.05	−579	40.0	0.566
		P−CDI	28.6	9.41	29.4	7.09	−0.809	2.59	−0.312	35.2	0.757
		DC	38.2	6.79	40.0	8.50	0.245	2.36	0.104	39.4	0.918
		*Total*	96.9	13.8	99.2	11.7	−2.33	3.97	−0.588	37.6	0.560
	*HSQ-ASD*	SI	3.10	1.69	3.68	1.50	−0.577	0.495	−1.17	40.0	0.251
		DS	2.48	1.41	2.46	1.60	0.026	0.465	0.056	40.0	0.956
		*Total*	2.67	1.34	2.92	1.42	−0.258	0.427	−0.604	40.0	0.549

Developmental Quotient (DQ); Mean (*M*); Standard Deviation (*SD*); Standard Error (*SE*); Parental Stress Index/Short Form (PSI/SF); Home Situation Questionnaire-ASD Version (HSQ-ASD); Parental Distress (PD); Parent–Child Dysfunctional Interaction (P–CDI); Difficult Child (DC); * equal variance not assumed.

**Table 2 brainsci-10-00649-t002:** Multivariate tests for primary outcome variables.

Effect	*Wilks’ Λ*	*F*(2, 33)	*p*-Value	*η* _p_ ^2^	Obs. Power
PSI/SF	0.293	39.9	<0.001 *	0.707	1.00
HSQ-ASD	0.821	3.59	0.039 *	0.179	0.624
Group	0.586	11.7	<0.001 *	0.414	0.990
Parental Gender (PG)	0.976	0.406	0.669	0.024	0.110
Group × PG	0.978	0.378	0.688	0.022	0.106
Intercept	0.830	3.37	0.046 *	0.170	0.596
Age	0.955	0.785	0.464	0.045	0.172
DQ	0.994	0.103	0.902	0.006	0.064

Developmental Quotient (DQ) i.e., Griffiths Mental Development Scales III total score; Parental Stress Index/Short Form (PSI/SF); Home Situation Questionnaire-ASD Version (HSQ-ASD); Design: Intercept + Age + DQ +PSI + HSQ + Group + PG + Group × PG. * *p* < 0.001.

**Table 3 brainsci-10-00649-t003:** Univariate Analysis for primary outcome variables.

Source	Dependent Variable	Hyp. df	df Errors	*MS*	*F*	*p*-Value	*η* _p_ ^2^	Obs. Power
Group	PSI/SF	1	34	686	18.7	<0.001 *	0.355	0.988
	HSQ-ASD	1	34	5.50	4.30	0.046 *	0.112	0.522
Parental Gender	PSI/SF	1	34	30.2	0.824	0.370	0.024	0.143
	HSQ-ASD	1	34	0.010	0.005	0.946	0.000	0.051
Group × PG	PSI/SF	1	34	11.1	0.305	0.586	0.009	0.083
	HSQ-ASD	1	34	0.660	0.517	0.477	0.015	0.108

Mean Square (*MS*). Design: Intercept + Age + DQ +PSI + HSQ + Group + Parental Gender + Group. * Parental Gender.

**Table 4 brainsci-10-00649-t004:** Pre- and postphase III comparison of outcome measures for control group (CG) and tele-assisted group (TG) separately.

Group	Test	Factor	*MD*	*SE*	*t*	df	BCa 95% Difference C.I.			*p*-Value
CG	PSI/SF	PD	−0.450	2.03	−0.225	19	−4.46	-	3.26	0.844
		P−CDI	1.65	2.13	0.777	19	−3.27	-	5.89	0.464
		DC	−3.00	2.34	−1.30	19	−7.45	-	1.79	0.225
		*Total*	−1.80	1.44	−1.19	19	−4.53	-	0.797	0.241
	HSQ-ASD	SI	0.050	0.172	0.288	19	−0.230	-	0.422	0.778
		DS	0.140	0.227	0.596	19	−0.250	-	0.573	0.542
		*Total*	0.020	0.133	0.145	19	−0.208	-	0.275	0.890
TG	PSI/SF	PD	3.13	1.75	1.80	21	−0.176	-	6.16	0.090
		P−CDI	3.68	1.22	3.02	21	1.17	-	6.32	0.010 *
		DC	0.227	2.50	0.091	21	−5.39	-	5.26	0.919
		*Total*	6.95	1.34	5.10	21	4.57	-	9.67	0.001 *
	HSQ-ASD	SI	0.823	0.393	2.06	21	0.036	-	1.53	0.052
		DS	1.03	0.351	2.90	21	0.378	-	1.69	0.012 *
		*Total*	0.863	0.367	2.32	21	0.137	-	1.57	0.035 *

Mean Difference (*MD*) between pre- and postintervention for each scale; Standard Deviation (*SD*); Standard Error (*SE*); Parental Stress Index/Short Form (PSI/SF); Home Situation Questionnaire-ASD Version (HSQ-ASD); Parental Distress (PD); Parent–Child Dysfunctional Interaction (P–CDI); Difficult Child (DC); Bias Corrected Accelerated Bootstrap (BCa). * *p* < 0.05.

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
