# Peer review of "Tele-Assisted Behavioral Intervention for Families with Children with Autism Spectrum Disorders: A Randomized Control Trial"

_brainsci, 2020, doi:10.3390/brainsci10090649_

Round 1

Reviewer 1 Report

brainsci-925745

The authors compared tele-assisted with in-person parent coaching in a sample of parents of children with ASD. They found relative benefits in tele-assisted coaching regarding parental stress and management of children’s behavior.

This is an interesting and well-written paper, that taps into a timely societal challenge: tele-services. However, I think the authors are not too clear about what is being evaluated (goals and program), and – maybe as a consequence - that they do not highlight optimally the significance of their results. In addition, I found the report of the results a bit confusing.

Goals and program

There is some ambiguity on what is being evaluated. In lns 71-73, the authors state that they aim to compare the efficacy of parent mediated interventions delivered by tele-health and in person with interventions delivered in-person only. Later, in lns 159-162 (program), they state something different: in phase III (the only phase that differs across groups) what is being compared is tele-assisted behavioral parent-training in the EG vs. in-person behavioral parent-training in the CG. Please clarify.

Program and scientific significance

Based on the latter description of the intervention program (lns 159-162), I would say that the critical comparison in this study relates to the medium that is used to deliver instructions to parents, since nothing else was different (or was it? Was tele-assistance more flexible in terms of schedule? Did it respond faster to parents’ needs? If so, please clarify).

That being so, I would say that the introduction should focus on the aspects of the different media (tele-assistance vs. in person) that may or may not help parents (rather than on parent-mediated strategies, which are common to both groups), and the hypotheses should be designed accordingly. For instance, did the authors expect tele-assistance to work better. If so, why?

This reasoning extends to the results and conclusions. Tele-assistance worked better than in-person assistance. Why was that so? Parents had more resources? Responses were faster? They had constant access to recorded instructions? They wasted less time? Of course, the authors will not be able to know this, but this can be discussed, at least in the introduction.

Analysis

Tables 3 and 4 (and the corresponding text in the results section) are confuse. With a MANCOVA, I was expecting to see first the multivariate analyses (effect of independent plus covariates on both dependent measures taken together), followed by ANCOVAs focusing on each dependent variable at a time. Instead, table 3 refers to effects from PSI and HSQ, which are dependent variables (could it be pre-PSI and pre-HSQ?), and table 4 refers to between-subject effects (was not table 3 about between-subjects effects too?). Please clarify.

T tests between pre- and post-measures for each group are the first result presented. I would say these tests should be more like post-hocs tests, and thus should follow the MANCOVA/ANCOVAs analysis. Moreover, if the authors are controlling for age and DQ (covariates), why is this ignored by the moment they move on with t-tests?

Sample size

The sample size seems small (20/22). I wonder if the authors made any a priori power calculations.

Author Response

brainsci-925745

The authors compared tele-assisted with in-person parent coaching in a sample of parents of children with ASD. They found relative benefits in tele-assisted coaching regarding parental stress and management of children’s behavior.

This is an interesting and well-written paper, that taps into a timely societal challenge: tele-services. However, I think the authors are not too clear about what is being evaluated (goals and program), and – maybe as a consequence - that they do not highlight optimally the significance of their results. In addition, I found the report of the results a bit confusing.

Goals and program

There is some ambiguity on what is being evaluated. In lns 71-73, the authors state that they aim to compare the efficacy of parent mediated interventions delivered by tele-health and in person with interventions delivered in-person only. Later, in lns 159-162 (program), they state something different: in phase III (the only phase that differs across groups) what is being compared is tele-assisted behavioral parent-training in the EG vs. in-person behavioral parent-training in the CG. Please clarify.

R. We thanks the reviewer for point out the incongruence, both groups received the same “in person” training during phase I and II, but actually the comparison was only for phase III, therefore we changed lines 71-73 into: “the efficacy of the parent-mediated intervention that was delivered via telehealth or in-person approaches”. We also added for clarity “The present study reports the results of phase III comparison between CG and TG groups.” In the 2.4 section lines 158-159.

Program and scientific significance

Based on the latter description of the intervention program (lns 159-162), I would say that the critical comparison in this study relates to the medium that is used to deliver instructions to parents, since nothing else was different (or was it? Was tele-assistance more flexible in terms of schedule? Did it respond faster to parents’ needs? If so, please clarify).

That being so, I would say that the introduction should focus on the aspects of the different media (tele-assistance vs. in person) that may or may not help parents (rather than on parent-mediated strategies, which are common to both groups), and the hypotheses should be designed accordingly. For instance, did the authors expect tele-assistance to work better. If so, why?

This reasoning extends to the results and conclusions. Tele-assistance worked better than in-person assistance. Why was that so? Parents had more resources? Responses were faster? They had constant access to recorded instructions? They wasted less time? Of course, the authors will not be able to know this, but this can be discussed, at least in the introduction.

R. We agree with the reviewer, unfortunately we only toke qualitative (informal) interviews with the participants. Nevertheless, we added a clarification in the discussion: lines 327-344, the introduction: 45-50 and the hypotheses: lines 78-81.

Analysis

Tables 3 and 4 (and the corresponding text in the results section) are confuse. With a MANCOVA, I was expecting to see first the multivariate analyses (effect of independent plus covariates on both dependent measures taken together), followed by ANCOVAs focusing on each dependent variable at a time. Instead, table 3 refers to effects from PSI and HSQ, which are dependent variables (could it be pre-PSI and pre-HSQ?), and table 4 refers to between-subject effects (was not table 3 about between-subjects effects too?). Please clarify.

T tests between pre- and post-measures for each group are the first result presented. I would say these tests should be more like post-hocs tests, and thus should follow the MANCOVA/ANCOVAs analysis. Moreover, if the authors are controlling for age and DQ (covariates), why is this ignored by the moment they move on with t-tests?

R. We thank the reviewer for the comment, something went missing in successive internal reviews of the paper among the different authors. We reviewed the presentation of the analysis to add clarity. Table 2 is now table 4 (table 3 is table 2 and table 4 is table 3 and reports univariate measures).

R. Univariate analyses and post-hoc comparison among groups were missing and we added them (lines 246-248).

R. The old table 2, new table 4, is a secondary analysis to present a pre-post change for each group (separately) while the MANCOVA assessed the differences between groups. We removed the covariate of DQ and age when moving to the single group analysis because we found no effect of DQ or age in the previous analyses.

Sample size

The sample size seems small (20/22). I wonder if the authors made any a priori power calculations.

R. Yes, we did, we now added it to the methods section: “Sensitivity analysis was performed using G*Power 3.1 for MANOVA with 2 groups and 2 factors and 2 covariates. With a total sample size of 42 we can identify large effects with sizes with f2(U) >.631.”

Reviewer 2 Report

This manuscript presents a relevant topic to publish in Brain Science, which could be accepted with some minor revisions. 

In my opinion, the introduction provides adequate information and structure to set up the research questions raised in manuscript; the methodology provides sufficient detail, but that can still be an improvement; results section is sufficiently clear and precise; the discussion of results based on previous literature should be more evident.

After carefully reading your manuscript, I point out some aspects that must be improved and corrected:

  • Some aspects of formatting should be corrected (spelling). Please, correct what it is pointed out in the body of the manuscript;

  • The abbreviation “TG” is used throughout the manuscript; however, the first time it appears coded as "intervention group" (line 134) . Later in the legend of table 2 is coded as "TG=Tele-assisted Group". Abbreviations should be defined in parentheses the first time they appear in the abstract, main text, and in figure or table captions and used

  • When reporting on research that involves human subjects, human material, human tissues, or human data, authors must declare that the investigations were carried out following the rules of the Declaration of Helsinki of 1975 (https://www.wma.net/what-we-do/medical-ethics/declaration-of-helsinki/), revised in 2013.  Example of an ethical statement: "All subjects gave their informed consent for inclusion before they participated in the study. The study was conducted in accordance with the Declaration of Helsinki, and the protocol was approved by the Ethics Committee of XXX (Project identification code)."

  • In the description of the outcomes measures, the authors should report a brief summary of their psychometric properties.  In addition, they should also report evidence concerning the reliability of the data collected;

  • All statistical symbols must be in italics (N, n, p, r, F ....). 

  • The discussion of results based on previous literature should be more evident. The authors write “…This is in line with the previous literature reporting how effective….” What are the studies?

Author Response

This manuscript presents a relevant topic to publish in Brain Science, which could be accepted with some minor revisions. 

In my opinion, the introduction provides adequate information and structure to set up the research questions raised in manuscript; the methodology provides sufficient detail, but that can still be an improvement; results section is sufficiently clear and precise; the discussion of results based on previous literature should be more evident.

After carefully reading your manuscript, I point out some aspects that must be improved and corrected:

  • Some aspects of formatting should be corrected (spelling). Please, correct what it is pointed out in the body of the manuscript;

we checked the text and had it reviewed by a native speaker.

  • The abbreviation “TG” is used throughout the manuscript; however, the first time it appears coded as "intervention group" (line 134). Later in the legend of table 2 is coded as "TG=Tele-assisted Group". Abbreviations should be defined in parentheses the first time they appear in the abstract, main text, and in figure or table captions and used

We changed abstract, main text, figures and table captions accordingly.

  • When reporting on research that involves human subjects, human material, human tissues, or human data, authors must declare that the investigations were carried out following the rules of the Declaration of Helsinki of 1975 (https://www.wma.net/what-we-do/medical-ethics/declaration-of-helsinki/), revised in 2013.  Example of an ethical statement: "All subjects gave their informed consent for inclusion before they participated in the study. The study was conducted in accordance with the Declaration of Helsinki, and the protocol was approved by the Ethics Committee of XXX (Project identification code)."

We changed paragraph 2.5 accordingly.

  •  

  • In the description of the outcomes measures, the authors should report a brief summary of their psychometric properties.  In addition, they should also report evidence concerning the reliability of the data collected;

We added the psychometric properties in the outcome measures descriptions (par. 2.6.1 and 2.6.2)

  • All statistical symbols must be in italics (N, n, p, r, F ....). 

 We changed abstract, main text, figures and table captions accordingly.

  • The discussion of results based on previous literature should be more evident. The authors write “…This is in line with the previous literature reporting how effective….” What are the studies?

We expanded the discussion and cited 4 previous studies. We also added a paragraph to the introduction.